# Opinion Polarization in Human Communities Can Emerge as a Natural Consequence of Beliefs Being Interrelated

**DOI:** 10.3390/e24091320

**Published:** 2022-09-19

**Authors:** Anna Zafeiris

**Affiliations:** 1MTA-ELTE Statistical and Biological Physics Research Group, Pázmány Péter Stny. 1/A, 1117 Budapest, Hungary; anna.kinga.zafeiris@ttk.elte.hu; 2MTA-ELTE ‘Lendület’ Collective Behaviour Research Group, Hungarian Academy of Sciences, Eötvös University, 1117 Budapest, Hungary

**Keywords:** belief systems, opinion polarization, belief system dynamics, fake news, cognitive biases, opinion dynamics

## Abstract

The emergence of opinion polarization within human communities—the phenomenon that individuals within a society tend to develop conflicting attitudes related to the greatest diversity of topics—has been a focus of interest for decades, both from theoretical and modelling points of view. Regarding modelling attempts, an entire scientific field—opinion dynamics—has emerged in order to study this and related phenomena. Within this framework, agents’ opinions are usually represented by a scalar value which undergoes modification due to interaction with other agents. Under certain conditions, these models are able to reproduce polarization—a state increasingly familiar to our everyday experience. In the present paper, an alternative explanation is suggested along with its corresponding model. More specifically, we demonstrate that by incorporating the following two well-known human characteristics into the representation of agents: (1) in the human brain beliefs are interconnected, and (2) people strive to maintain a coherent belief system; polarization immediately occurs under exposure to news and information. Furthermore, the model accounts for the proliferation of fake news, and shows how opinion polarization is related to various cognitive biases.

## 1. Introduction

As Evan Williams, co-founder of Twitter, famously said in a 2017 interview, “I thought once everybody could speak freely and exchange information and ideas, the world is automatically going to be a better place. I was wrong about that” [1]. This is a good overview of the surprise that the unprecedented connectivity among people—primarily driven by various internet-based social media platforms in the early 21st century—brought us unprecedented factions, dissension and fake news [2,3], instead of agreement and conciliation [4,5]. The reasons behind these phenomena are diverse and manifold and, accordingly, are the subject of the most diverse scientific fields from history [6] through sociology [7] to computational social science [8,9,10].

The quest for finding a valid explanation—and a practicable model—for the phenomena of the above mentioned polarization and fragmentation has been underway for decades. Specifically, regarding computational models, an entire field, *opinion dynamics* , has emerged in order to study the way opinions, information, views and beliefs propagate in human communities [11,12,13,14,15,16]. In general, these models study the dynamics of attitudes related to a certain topic, such as issues related to climate change, abortion, immigration, vaccination, a certain politician or political party, etc. Typically, the attitudes of the agents towards the given issue are described by scalar values which are assumed to be altered due to communication with peers [11,12,17]. Within this scientific field, *consensus* refers to the state in which all social actors share the same opinion, *polarization* to the condition when each agent accepts one of two opposing opinions, while *fragmentation* refers to a transitory phase, in which a finite number of distinct “opinion islands” appear.

A key concept related to continuous opinion dynamics models—models representing opinions as scalars taken from a continuous interval usually between −1 and +1—is the “confidence threshold”, which is a value above which agents cease to communicate with each other. Above this threshold, “classical” bounded confidence models predict consensus, and fragmentation and/or polarization under it [13,14,15,17]. However, the polarization/fragmentation breaks down in the presence of noise [18,19,20,21,22]—an essential and unavoidable component in all social and biological systems [17]. In order to reach polarization, other mechanisms have been suggested, such as “distancing”, which is the direct amplification of differences between dissimilar individuals [18]. Lately, these models have been further developed by incorporating more realistic features, such as heterogeneity with respect to the agents’ confidence thresholds [23] or by incorporating an underlying communication network [24] defining the pattern by which social actors interact with each other. These approaches lead to more complex—and realistic—dynamics. Furthermore, recently, building on the observation that *events* often have a polarization effect [25,26], the process of polarization was modelled by extending a classical bounded confidence model—the so called Hegselmann-Krause model [13]—in a way that individuals change their opinions in line with the certain event [27]. Other models approach the problem from a kinetic [28] or hydrodynamic [29] perspective.

In the present paper, an alternative, or complementary, explanation is suggested, by showing that in case some “basic” human characteristics are incorporated, polarization immediately appears once agents are exposed to new information—even without direct communication. These almost trivial human characteristics are that (i) human beliefs are interrelated rather then evolving independently of each other, and (ii) people strive to maintain a coherent, contradiction-free belief-system. The fundamental difference between “classical” opinion dynamics models and the ones incorporating such features is that while in the first case agents are represented with a single scalar value in the second case the model of the social actors has some kind of inner structure. This difference gives rise to an entirely different dynamic [30,31].

## 2. The Background: Fundamental Features of Human Belief Systems

During the last decades, a vast amount of knowledge has accumulated in scientific fields on the ways humans perceive the world, make decisions and structure their beliefs. Despite the fact that obtaining a detailed understanding of these processes still require further work, some scientific fields, such as neurobiology [32,33,34,35,36] or various human sciences [37,38], such as psychology [39,40,41,42], anthropology [43], economics [44,45,46] and political science [4,5,47,48,49,50,51] have progressed considerably. From our point of view, the key finding is that in humans, opinions and beliefs *never* occur alone, that is, no concept or belief can exist in isolation. (Actually, humans are not even able to memorize anything without connecting it to something meaningful [32].) Rather, concepts and beliefs are organized into a structure, a system (*"belief system"*) which has well-defined features [32,47,48,52]:*First* and fore-most, it seek to be *consistent*. This means that people try to maintain a belief system in which the elements mutually support each other, or are independent [48,52]. In the case of holding *conflicting* beliefs, people experience discomfort called *cognitive dissonance* [53,54] which people will try to reduce. By this time, cognitive dissonance has become one of the most influential and well-researched theories in social psychology [54,55,56].*Secondly*, beliefs are not equally important: those that are more personal, closer to the “self”, “identity” or “ego”, trigger more intense feelings and are more difficult to change [43,46,54,57]. In this sense, beliefs have a *hierarchical property*: the ones that are higher in rank define or constrain the ones that are lower in rank [58]. For example, the belief or disbelief in God is a central (high-ranking) element in one’s belief system, while the belief that “an egg should be boiled for seven minutes in order to get the best soft-boiled egg” is a low ranking one, and, accordingly, can be changed more easily [43]. Furthermore, beliefs that people hold in high regard tend to cause greater dissonance in case of contradiction with other beliefs [54].*Finally*, beliefs belonging to the same broader topic (e.g., health, art-related topics, religion, political issues, etc.) are more strongly interrelated than beliefs belonging to different topics. For example, attitudes towards “freedom of speech”, “religious freedom” and “freedom to choose spouse” are more closely related than beliefs regarding “freedom of speech” and, say, homeopathic treatments. In mathematical (graph-theoretical) terms, belief systems are “modular” or “compartmentalized” [43,50].

As a first approximation, such a structure can be represented as a *modular*, *hierarchical network* (graph) in which the nodes are connected by supportive (positive) or invalidating (negative) relations [48,49,50,59] (Figure 1). For example, according to a widespread belief, “Pathogens can cause diseases”. According to another wide-spread belief—primarily in historical societies—“Diseases are caused by evil spirits” [60,61]. These two concepts are in negative relation: somebody believing in one of these opinions will probably disagree with the other. In contrast, the beliefs “Pathogens can cause diseases” and “Contagion is due to the spread of pathogens” support each other (positive relationship), since accepting one of them renders the acceptance of the other more probable.

This last requirement ensures that the *level of consistency* can be defined [62]. In this description, nodes are beliefs, and edges represent functional relationships between them [52,59,63]. This approach has already been applied by sociologists and economists as well in order to model political belief system dynamics [49,50,59,63,64]. Within this framework, the focus is on the *relatedness* of beliefs—captured by graph representation—while the hierarchical and modular characteristics do not gain special importance (see the Appendix A for the results incorporating the hierarchical characteristics as well).

In these models, a node is “an element of a person’s belief system” [59], which the related literature names slightly differently, such as “opinions” [49], “concepts” [62] “attitudes” [49,59,64], “beliefs” [50,59,62], or “positions” [63,64]. In the present article, the terms *“belief”*, *“attitude”* and *“concept”* are mostly used, under the condition that we consider all sorts of human thoughts as an “element of a belief system” (that is: a “belief”), whether they be simple or complex, that can be transmitted with the use of language from one person’s mind to another’s [38,43]. Accordingly, a node can be a thought or attitude towards public issues such as abortion, climate change, immigration, vaccination, gun control; it can be information regarding a public figure or a political party, an idea related to “proper behaviour”, “justice”; or the belief that somebody did or said something, etc. Furthermore, this definition implies that beliefs can be transmitted via communication. Communication, in its most basic form, can be discussion (talking) between two or more individuals, but it can also be news/beliefs/information spread by a single agent or organization to many people at the same time, for example via public and social media, news channels, journals, etc. In short, communication is the circulation of news, information and beliefs within a certain community.

The fact that beliefs (attitudes/concepts) are in functional relation with each other (that is, if they are connected, they either support or contradict each other) is crucial, because it implies that some “new” belief will fit into an already existing system—the ones that *in*crease the system’s consistency—while others—the ones *de*creasing the system’s consistency—will not [52]. These latter gives rise to the disturbing feeling of cognitive dissonance and people will apply various strategies in order to avoid them. Among other strategies, they will try to keep contact only with those from whom they expect reassuring information (homophily), they will ignore certain information and focus on other information (attentional bias), while greater credence will be given to evidence that fits with the existing beliefs (confirmation bias). These strategies are known as various *biases* in the field of psychology and sociology [40,65,66,67]. Different people apply different strategies to various extents; however, to some level, all the strategies are applied by all of us [40,41,65,66].

Furthermore, these cognitive dissonance avoiding mechanisms are in close relation with the proliferation of fake news and the circulation of various types of questionable information as well. In case the well-fitting of a piece of information into the already existing belief system weights more than its credibility, people will adopt it—simply because it provides the pleasant feeling of reassurance. This mechanism is applied in all aspects of life, not only in case of political issues. For example, in the field of economics, it has been observed that managers, whose sales fall short of expectations, rather than rethinking the qualities of the product, tend to identify the cause of the failure elsewhere, for example in the marketing campaign. In such cases, they state that the *marketing* campaign failed, so it is actually a miracle that the product was sold at all [68]. By finding this new explanation, the sales results show directly the *merits* of the product, not its failure, and as such, serves as a basis for the pleasant feeling of reassurance. This mechanism is analyzed in Section 4.2.

## 3. The Model

We assume a population of *N* agents. At this point, only their attitudes towards two concepts are important (the attitudes towards, say, vaccination (Concept 1) and, say, a certain public figure (Concept 2)). People can hold any kind of attitudes towards these concepts, from total condemnation (marked by −1) to total support (denoted by +1). Neutrality or indifference is indicated by zero or near-zero values. We are interested in how the agents’ attitudes evolve due to being exposed to some news (piece of information) that creates a relation between two originally independent concepts (see Figure 1b).

The relation $K_{0}$ can be positive or negative. Using the above example, a trivial positive connection can be that “XY public figure (concept 2) has spoken out *in favor* of vaccination (concept 1)” ($K_{0}=+1$), while a negative connection can be that “XY public figure has spoken out *against* it” ($K_{0}=-1$).

In case an agent holds positive attitudes towards both concepts, the positive message (support of vaccination) will give rise to the comforting feeling of *reassurance* [41]. In this case, the original attitudes are reinforced, since both concepts become better connected and further embedded into the belief system. In contrast, in case the XY trusted and respected politician takes a position *against* vaccination, a supported matter, the agent will experience *cognitive dissonance* with an intensity proportional to the original attitude values, and will apply a strategy in order to reduce it [53,54].

Turning back to the basic scenarios, an agent can hold negative attitude towards one of the concepts, and a positive one towards the other—say, for example, a negative attitude towards the public figure and positive attitude towards vaccination. In this case, a negative relation will give rise to reassurance, i.e., “XY politician, whom I anyway hold very low, talked out against vaccination, a cause so important for me ... No surprise here, a fool is known by his conversation” and so on. All scenarios can be analyzed with the same train of thought.

Accordingly, from a mathematical point of view, the cognitive dissonance (or reassurance), $C_{i}\left(t\right)$, that agent *i* will experience at time-step *t*, can be formulated as:(1)$$C_{i}\left(t\right)=a_{i,1}\left(t\right)\cdot a_{i,2}\left(t\right)\cdot K_{0}\;$$
where $a_{i,1}\left(t\right)$ and $a_{i,2}\left(t\right)$ are the original attitudes of agent *i* towards concept 1 and 2, respectively, at time-step *t* , and $K_{0}$ is the type of connection, which can take two values, +1 or −1, according to the supportive or opposing nature of the connection between the concepts (see also Figure 1b). In case *C* is positive, it is called *reassurance*, while in case it is negative, it is usually referred to as *cognitive dissonance*. Anyhow, in both cases, *C* denotes the value by which the information alters the *coherence* or *consistency* level of agent *i*’s belief system. Note that $|C_{i}\left(t\right)|\leq 1$ is always the case, since $|a_{i,1}\left(t\right)|\leq 1$, $|a_{i,2}\left(t\right)|\leq 1$, and $K_{0}=\pm 1$.

According to the literature [40,41,65,66], in case of facing information inducing cognitive dissonance, people attempt to relieve the discomfort in different ways, among which the most common ones are:(i)Rejecting new information that conflicts with the already existing ones;(ii)Re-evaluating the attitudes;(iii)A tendency of “explaining things away”, that is, finding alternative explanations (developing new beliefs) which supplement the original information in a way that the primordial contradiction is dissolved.

From a modeling point of view, the first strategy—rejecting the information—simply leaves the belief system unaltered. In this case, in the framework of the model, the network—nodes, edges and weights—remain unchanged. The second and third strategies do modify the belief system, due to the new connection between the originally unconnected concepts. In the following section, we will focus on modelling these strategies.

### 3.1. Modelling the Re-Evaluation of Beliefs

The constant re-evaluation of our already existing beliefs is an inevitable part of the process of learning and development [32]. New information often comes in the form of creating connection among concepts and beliefs that were originally disconnected. As a matter of fact, this is a basic form of learning. Furthermore, people tend to evaluate most information, beliefs and concepts according to some personal narrative, a personal “frame of mind”, which is different from person to person. Simply put, this unique narrative is our personality [37], which defines the very way we perceive the world and make decisions [45,46]. This variety entails individual differences in evaluating the most diverse topics around us, whether it be the judgement of a public figure, a movie or the question of immigration.

In the context of a formal model, the most simple and plausible way to grasp these attitudes is to use numbers between −1 and +1 in a way that negative values represent negative attitudes and positive ones refer to positive stances. The two extreme values, −1 and +1, refer to complete condemnation/approval, respectively.

In order to see *how* these values might change, consider for example the following case: Paul believes that, say, genetically modified food is harmful. He has already heard it from his friends, and now he reads it in his favorite blog as well. This gives him a feeling of reassurance, due to which he will be convinced about the verity of this belief even more, and will be more attached to his favorite blog as well. In other words, the "embeddedness" of the original attitudes will increase. Mathematically speaking, his already positive attitudes towards these concepts (his belief and the blog) will increase even more due to the positive connection. Now consider a situation where he learns the opposite from his favorite blog, namely that there is nothing at all that could be harmful in genetically modified food (that is, a *negative* association appears among the two positive concepts: the belief and the blog). In this case, he will experience some level of cognitive dissonance, whose extent depends on his original commitments towards the two concepts [42,53,56]. This experience will make him less convinced, either of the reliability of the blog or of the belief itself—or both. Mathematically speaking, the originally positive values (attached to the two concepts) will decrease somewhat. In other words, *cognitive dissonance* (negative $C_{i}\left(t\right)$ values) *de*creases the absolute value of the affected attitude (*k*), while *reassurance* (positive *C* values) *in*creases it. Consider the following formula:(2)$$a_{i,k}\left(t+1\right)=sign\left(a_{i,k}\left(t\right)\right)\cdot \left(|a_{i,k}\left(t\right)|+\rho \cdot C_{i}\left(t\right)\right)+Z_{A}$$
where $a_{i,k}\left(t\right)$ is the original attitude of agent *i* at time-step *t* towards attitude *k*, $sign(a_{i,k}\left(t\right))$ is its signal (+ or −1), ρ is a random value (“noise”) taken from the [0,1] interval with uniform distribution, effecting the extent to which the attitude changes, and $C_{i}\left(t\right)$ (defined by Equation (Equation 1)), is the level of “coherence” (commonly known as *cognitive dissonance*, in case it is negative, and *reassurance* in case it is positive). Finally, the $Z_{A}$ noise comprises the effects of other factors influencing the change of attitudes. It can be either positive or negative with equal probability. In case the updated attitude value $a_{i,k}\left(t+1\right)$ falls outside the predefined [−1,1] interval, it is set to the nearest threshold (+1 or −1).

Attitudes do not vary with the same probability and to the same extent in case of different people and topics; for some, environmental issues are extremely important (and “nothing can change” this stance), some people are detached, while others are convinced that they are just evil-minded hoaxes. The more extreme an attitude is, the more difficult is to change it [43,46,54,57]. (See also Section 2, 2nd bulleted point, “hierarchical property” of belief systems).

Mathematically speaking, the feature *"more difficult to change"* can be introduced into the model in two ways:The more extreme an attitude value *a* is, the lower the *probability* that it will change. Equation (Equation 3) expresses the most simple mathematical formulation of this relation.The more extreme an attitude value *a* is, the smaller the *magnitude* with which it can change.

For the results presented in the main text, the above mentioned hierarchical property was introduced into the model according to the first way, that is, by setting the *probability* $p(AttCh_{i,k}\left(t\right))$ of attitude-change according to Equation (Equation 3), and setting ρ—the parameter controlling the maximal *extent* with which the attitude values alter due to the experienced cognitive dissonance or reassurance (*C*)—to 1. In other words, in Equation (Equation 2), ρ=1 for the results presented in the main text. In the Appendix A, a detailed analysis is provided on how the parameter ρ effects the simulations (leading to the conclusions that the main claims remain valid, independently of the maximal extent of the alterations, see Appendix A).
(3)$$p\left(AttCh_{i,k}\left(t\right)\right)=1-\left|a_{i,k}\left(t\right)\right|$$

Note that in all the equations, the updated attitude values depend only on the agents’ previous attitude values ($a_{i,k}\left(t\right)$), the type of the news ($K_{0}$), and on the cognitive dissonance (or reassurance) values that the news creates in the agents ($C_{i}\left(t\right)$). This means that agents develop their attitudes independently from each other. This originates from the fact that *the source of the information does not matter* in the present model. Accordingly, if the assumption is that it is the *agents* who circulate the news among themselves (for example in the form of “gossiping” either in person or on social media), then they interact with each other. In contrast, if the source of the information is something else (for example, state media or some kind of propaganda) then agents do not interact directly with each other. In reality, information usually circulates in both ways. The reason why entire populations are considered is twofold. Firstly, because one single agent cannot *“polarize”*; they can develop extreme attitudes under certain circumstances. *Polarization* is an emergent, statistical property of communities, a phenomenon which does not have an interpretation on the level of individuals. The larger the statistics, the more apparent the phenomenon. The second reason is that after studying the elementary process of attitude-update in detail (which is the topic of the present paper), an immediate next step is to study the way by which agents manipulate and organize their social ties (links) assuming similar motivations (avoiding cognitive dissonance and enjoying reassurance).

### 3.2. Modelling the Inclusion of New Beliefs in Order to Relieve Cognitive Dissonance

In case a social actor experiences the upsetting feeling of cognitive dissonance due to a certain piece of information, a commonly applied strategy is to adopt—or create—an even newer belief that changes the context of the original one in a way that it does not serve as a basis of cognitive dissonance any longer; rather, it becomes neutral or even gives rise to the pleasant feeling of reassurance [40,41,68]. An example of this maneuver is mentioned at the end of Section 2, related to managers whose sales data lag behind the expectations tendentiously conceive of various explanations, e.g., ones related to “awfully managed” marketing campaigns. By adopting this new belief (namely that the marketing campaign was awfully managed), the cognitive dissonance caused by the negative sales results (linking a failure to their “self”) is eliminated; furthermore, in this light, the sales-results could be seen as an achievement rather than a failure.

The most simple assumption is that the probability $p(NB_{i}\left(t\right))$ of adopting such a new belief (by agent *i* at time-step *t*) is proportional to the relief its adoption provides. Since only positive $C_{i}^{NB}\left(t\right)$ values represent reassurance, the most simple mathematical formula is the following:(4)$$p\left(NB_{i}\left(t\right)\right)=max\left(0,C_{i}^{NB}\left(t\right)\right)$$
where,
(5)$$C_{i}^{NB}\left(t\right)=a_{i,1}\left(t\right)\cdot a_{i,3}\left(t\right)\cdot K_{NB}\;$$

As shown before, $a_{i,1}\left(t\right)$ and $a_{i,3}\left(t\right)$ are the attitudes of agent *i* towards concept 1 and the new belief at time-step *t*, respectively, and $K_{NB}$ is the (positive or negative) connection type between them. Note that in case $C_{i}^{NB}\left(t\right)$ is negative—marking cognitive dissonance, instead of reassurance—the agent is highly unlikely to adopt the new belief.

## 4. Results

### 4.1. Re-Evaluating Beliefs

Let us consider a population in which the agents’ initial attitudes towards two arbitrarily chosen concepts are distributed uniformly, taking values from the [−1,1] interval. In other words, at the beginning of the simulation, all sorts of attitudes are present in the population with equal probability, from complete condemnation to complete support and everything in between, with an average of zero. Let us now assume that this population is exposed to some kind of news, connecting the two originally unconnected concepts.

Assuming the most general setup, at each time-step *t*, a randomly chosen agent *i* acquires the information, and updates his/her attitudes according to Equation (Equation 2). (For the flowchart of the algorithm, see Figure A1a. The source of information can be anything, such as public or social media, propaganda, government information, etc. As it can be seen in Figure 2a,b,d,e, proportionally to the level of exposure (iteration number *t*), the attitudes tend to move towards the two extreme values, +1 and −1, either due to the experienced reassurance or due to the attempt to reduce cognitive dissonance (Equation (Equation 2)). The *distribution* of the attitude values within the population evolves very similarly in case of the two attitudes, since both are governed by Equation (Equation 2). (See Figure 2a,b,d,e). At high iteration numbers (indicating strong exposure to the news), around half of the population fully *supports* Concept 1—marked by attitude values close to +1—while the other crowd—composed of those whose attitude values are close to −1—fully *rejects* it. In other words, the population is *polarized* with respect to Concept 1 (Figure 2a,d). The same applies to Concept 2 (Figure 2b,e). In case the type of connection ($K_{0}$) is negative (Figure 2 bottom row), the two “stable points”—adopted by the vast majority of the population—are (+1,−1) and (−1,+1), that is, where the two attitudes are reversed, either complete rejection of concept 1 and complete acceptance of concept 2 occurs, or vice versa. These are the two peaks in Figure 2f. In a symmetric manner, in case the connection type, $K_{0}$ is positive, the vast majority of the population will either completely support both concepts (one of the peaks will be at (+1,+1)), or will completely reject both of them (the other peak will be at (−1,−1)), as in Figure 2c. That is, independently of the type of connection, the originally uniformly distributed attitudes will tend towards the extremities, meaning that the mere attempt to maintain a consistent belief system alone promotes the processing of attitudes tending towards extremities in case of being exposed to persistent information. Of course, in reality, it is not only one type of news that circulates within a community, but many types, often with different messages and connotations, but it is certainly an important—and so far overlooked—point, that this human drive (the urge to maintain consistent beliefs) alone has the capacity to push attitudes towards extremities—a phenomenon increasingly experienced in our increasingly connected world.

In Figure 3c “extremity” is defined as “being closer to +1 or −1 than a certain threshold value ϵ”. Accordingly, if ϵ=0.01, then attitudes between 0.99 and 1, and attitudes between −0.99 and −1 will be considered as “extreme”. Similarly, if ϵ=0.1, then attitudes between 0.9 and 1, and the ones between −0.9 and −1 will be considered as "extreme". Apparently, as can be seen in Figure 3c, the exact value of ϵ does not matter.

Note the small peaks around near-zero values in Figure 2a,b,d,e, at small *t* values. According to the simulations, *in case of limited exposure to the news*, agents might also adopt neutral standpoints (marked by near-zero attitude values) in order to avoid cognitive dissonance. This phenomenon is highlighted in Figure 3b. However, this is an unstable equilibrium point, since any further information regarding the given concept (appearing as noise $Z_{A}$ in Equation (Equation 2)) pushes the attitude value away from zero. (See also Appendix A).

### 4.2. Finding Relief in New Ideas

As has already been mentioned, the other “basic strategy” applied by people in order to reduce the unpleasant feeling of cognitive dissonance is to reinterpret the incoming information by placing it into a context in which the contradiction vanishes, or even better, serves as a basis for reassurance [53]. For example, a doctor in his blog recollected memorable moments of the first year of the COVID-19 pandemic [69]. He remembers that when he tried to convince his family members to take the vaccine, he received vehement rejection, which was settled by receiving the comment that “You have good intentions, we know it. But you do not see the reality, because the “big players” leave you out from the party”. As it turned out, by this they meant that the “big players” know perfectly well that the pandemic is a hoax, but they use the everyday doctors—such as the one writing the blog—for their purposes, i.e., to force “everyday people” into take the unnecessary and harmful vaccine. In this example, the doctor is a positive concept in the eye of his relatives, but the epidemic is negative (considered to be a hoax). When it turned out that the doctor considered the epidemic real (hence they should take the vaccine), he created a positive (supportive) relation between himself and the pandemic. This resulted in cognitive dissonance in the relatives, which was dissolved by adopting the new belief (about the “party” of the “big players”), which allowed the original attitudes to remain unchanged.

In the context of the present framework, this scenario can be represented by supplementing the original graph (including two nodes and an edge between them, as in Figure 1) with a new node, representing the new belief (see Figure 4a. The new belief can be related to either of the original concepts, or to both of them. As an example, in Figure 4a, the new belief is connected to Concept 1. The type of connection, $K_{NB}$, can be either supportive or contradictory, similarly to the connection relating the two original concepts, $K_{0}$.

Figure 4b depicts the “stable configuration” which the dynamics tends towards. As has been shown already, in the case of $K_{0}=-1$, the attitudes towards concept 1 and 2 tend to be antagonistic and extreme (marked by the attitude values accumulating in the (−1,+1) and (+1,−1) points on the x−y plain), while in the case of $K_{0}=+1$ (positive relation), the attitudes towards concept 1 and 2 tend to be coincidental and also extreme (marked by the attitude values accumulating in the (+1,+1) and (−1,−1) points on the x−y plain, see Appendix A). The vertical, *z* axis depicts the attitude values towards the new, cognitive-dissonance-relieving belief; those who adopt it tend to develop an extreme relation towards this belief as well (in case of unceasing exposure). In contrast, those for whom the approval of the new belief would create cognitive dissonance, simply reject its adoption. In terms of the model, in their case, the edge $K_{NB}$ will simply not exist, and hence the node representing this belief will not be connected to the belief network. The two solid columns belong to these agents, depicting their original attitudes, which simply do not change throughout the simulation. (In case we stipulate that only those values are shown in the figure which participate in the belief system of an agent, these values could be omitted as well, but for sake of clarity, in Figure 4b, we have kept them.) Furthermore, a small vertical “cloud” can be seen in the middle of the chart, representing those who are neutral towards the original concepts, i.e., their attitude towards the new (cognitive-dissonance-relieving) belief can take any value. Importantly, as is apparent from Figure 4, this mechanism pushes the attitudes towards extremities as well.

## 5. Discussion

The detailed methods by which humans perceive and make sense of the world—despite some eminent achievements [33,34,38,39]—is still to be understood. However, some basic characteristics have been elucidated by now, and have become part of mainstream science as well [32,35,36]. One such characteristic is that in the human mind, beliefs are strongly interconnected, and as such, no belief, concept or “piece of information” can exist on its own. Furthermore, in case of new information, humans immediately attempt to interlock it in a coherent way, seeking for connections and support with already existing beliefs.

Accordingly, the novelty of the present model lies not in “assuming” the above-mentioned two human characteristics—since they are well-studied and widely accepted by main-stream science [43,53,55]—rather, it lies in their mathematical formulation and incorporation into agent-based models.

There are two more further points worthy of consideration related to the model:(i)Real belief systems have a tremendous amount of elements (instead of two or three), that are interconnected and embedded into each other in a complicated manner [39,43], and, accordingly, the “optimization process”—the attempt to minimize the contradictions among the components—refers to the *entire* system. From a physicist’s point of view, this process is in close relation to physical structures aiming to reach an energy minimum. In this approach, “different realities” [57] can be different local energy minimums of similar systems. However, it is imperative to understand the *elementary relation* between *two* elements of the system before considering the entire structure. The present manuscript focuses on this elementary relation. Graph representation is important because, and only because, it serves as a mathematical tool for handling interrelated entities (which are the “beliefs” or “concepts” in our case). Since in the human mind a vast amount of concepts and beliefs are interrelated densely and intricately, any of its graph representations must also assume a vast amount of intricately interrelated (linked) nodes. However, from the viewpoint of the present study, the specific type of the graph does not play any role, because we focus on the elementary process altering the characteristics of two nodes (namely the “attitude values”) due to a newly appearing link between them. (If a link appears, it is due to a certain piece of information connecting the two, originally unconnected beliefs/concepts). The nodes whose values alter are selected by the link (representing a piece of information).(ii)The present model does not assume that the repeated information is *exactly* the same, only that the *type of connection* between two concepts (say a political party and a public issue, such as immigration or environmental topics) is tenaciously either positive or negative. Hence, it also explains how attitudes can become extreme due to the continuous repetition of information, and as such, it serves as a complementary explanation [70] for the reason why, throughout history, the most diverse regimes found it useful to repeat the same messages over and over again (despite the fact that everybody had already heard them many times).

Furthermore, the present model has some additional results as well, which are yet to be studied. Specifically, according to the results, the attitudes a certain type of news or information triggers depend on the *intensity* of the exposure. More precisely, in case of limited exposure, people tend to develop a centralist attitude first (which is an unstable equilibrium point), which, in case of persistent news-circulation, give way to extreme stances. The dynamics under *limited* exposure to news was not studied extensively in the present manuscript.

The ambition of the paper was to call attention to certain human traits that have not yet been incorporated into current computational models aiming to simulate opinion dynamics in human communities. From this perspective, the main point is the naturalness by which polarization can emerge, despite the fact that the model incorporates only minimal assumptions which are considered to be part of well-established, main-stream scientific results.

## Figures and Tables

**Figure 1 entropy-24-01320-f001:**
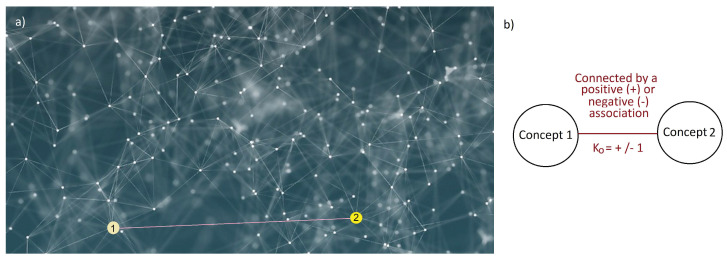
The graph representation of belief systems. (**a**) As a first approximation, human belief systems can be represented by networks in which nodes are beliefs (“elements of a belief system” [59]) and edges represent relationships. (**b**) Links (relationships) can be either supportive (positive) or contradictory (negative) [62].

**Figure 2 entropy-24-01320-f002:**
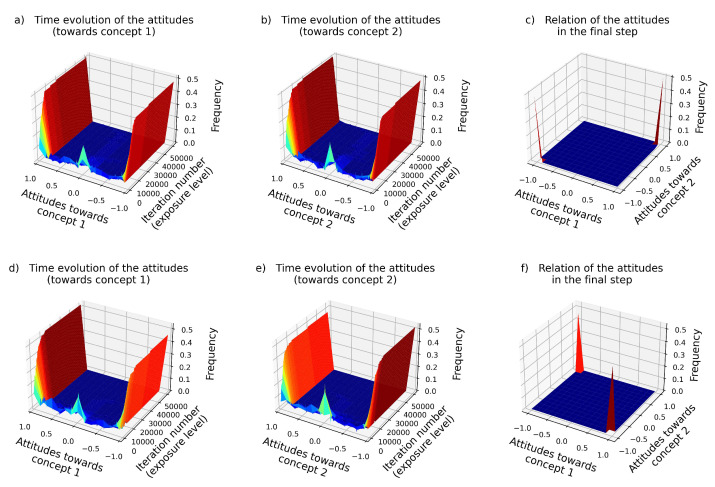
A typical time evolution of two attitudes ($a_{i,1}\left(t\right)$ and $a_{i,2}\left(t\right)$) within a population of N=100 agents. Top row: $K_{0}=+1$ (supportive relation). (**a**,**b**): the distribution of attitudes values towards concepts 1 and 2, respectively, as a function of time *t*. (**c**): At the final state, the vast majority either supports both beliefs (marked by the peak at (1,1)) or rejects it (marked by the peak at (−1,−1)). Bottom row (**d**–**f**): $K_{0}=-1$ (conflicting relation). (**f**) The major difference in this case is that at the end of the simulation most agents support one of the beliefs and disagree with the other (marked by the sharp peaks at the (+1,−1) and (−1,+1) points). The parameters are: population size N=100, number of iterations *T* = 50,000, and connection type $K_{0}=-1/+1$, and the noise value is $Z_{A}=0.01$.

**Figure 3 entropy-24-01320-f003:**
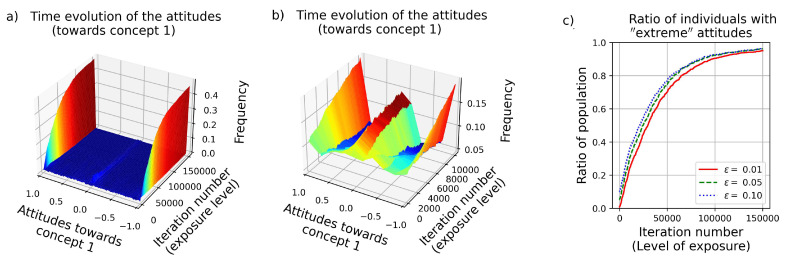
A typical time evolution of an attitude value $a_{i,k}\left(t\right)$ as a function of time (*t*) for (**a**) large population (N=1000), and (**b**) under limited exposure to news (*T* = 10,000). In this case, since in each time step 1 individual learns the news (out of the N=1000), on average, each agent will have heard it 10 times at the end of the simulation. As can be seen, for such a level of exposure, developing a neutral standpoint (adopting attitude values close to zero) is a good “strategy” as well. However, this neutrality vanishes in case of more enduring circulation of the news. (**c**) Proportional to the level of exposure (iteration number *t*), the ratio of the population holding “extreme” attitudes monotonically grows, independently of how “extremity” is defined (by the parameter ϵ). The parameters are: population size N=1000, Number of iterations *T* = 150,000 (except for sub-figure (**b**), on which *T* = 10,000), connection type between the concepts $K_{0}=-1$ and the noise value is $Z_{A}=0.01$.

**Figure 4 entropy-24-01320-f004:**
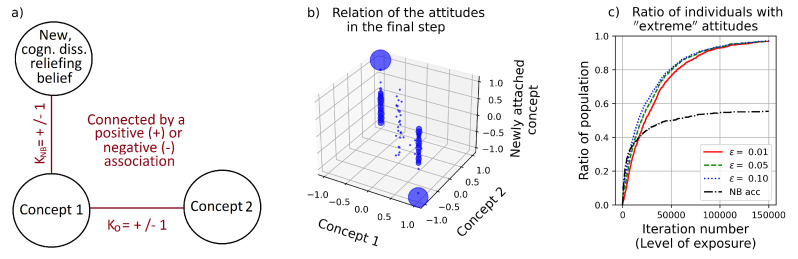
Change of attitudes within a population due to the circulation of some news, connecting concepts 1 and 2. In this case, agents *might* adopt a new belief as well, in case it reduces their cognitive dissonance. (**a**) The new belief can be connected to either or both concepts 1 and 2. (**b**) The “stable configuration” toward which the dynamics tends to (after 150,000 simulation steps). (**c**) The ratio of individuals holding “extreme beliefs”, and adopting the new belief (black semi-dotted line). “Extreme attitudes” are those closer to +1 or −1 than a certain threshold value ϵ, such as 0.01, 0.05 and 0.1. As it can be seen, the ratio is largely independent of the exact value of ϵ. The parameters are: N=1000, *T* = 150,000, $K_{0}=-1$, $K_{NB}=-1$ and $Z_{A}=0.01$.

## Data Availability

The source code of the simulation can be found on CoMSES, a public computational model library, under the following link: https://www.comses.net/codebases/2ccbae5e-d51c-41ed-80c2-5e4b33c4e9c0/releases/1.0.0/ (Accessed on: 13 September 2022).

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
