# Peer review of "Opinion Polarization in Human Communities Can Emerge as a Natural Consequence of Beliefs Being Interrelated"

_entropy, 2022, doi:10.3390/e24091320_

Round 1

Reviewer 2 Report

I must say this is a very refreshing paper, that I thoroughly enjoyed reading. I strongly believe that too many models of opinion dynamics just rely on adding a small tweak to existing models, and that such tweaks are usually disconnected from reality. Instead, this paper proposes a new model heavily based on evidence from cognitive science, which is fundamental to understand how opinions form. Overall, this is a clearly written and well-structured paper, and most of my comments are really just out of curiosity.

I) In equation 2, the author introduces a random variable, rho, to model the "quenching" of attitude changes. Why is this necessary and why choosing a value taken from a uniform distribution?

II) In equation 3, the author introduces a mechanism thereby agents update their opinion with a higher probability the closer their current opinion is to being neutral (ie 0). I don't know if there is any research to back this up, but anecdotally I would say that people who do not have an opinion on something whatsoever, tend to remain generally apathetic or uninterested about that specific topic. Wouldn't it be more reasonable to have then a bimodal function in which 0 is also considered an extreme?

III) This is a question I always ask when reviewing papers on opinion dynamics, and I believe it is still applicable here even though the author draws extensively from existing empirical evidence in social and cognitive sciences: how do you validate your model? How can you tell that the model is good if you do not use any empirical data to validate your results? Input validation is great, but output validation is also required to make sure a model works properly. I appreciate that the goal of the paper is to start a conversation rather than provide conclusive results, but I feel that output validation still needs to be at least properly discussed. 

IV) there are a few typos here and there. These are the ones I caught:

line 8 in figure 2. Should be "is" instead of "are"

line 368: "refer" should be "refers"; "physicist" should be "physicist's"

Reviewer 3 Report

see attached file

Round 2

Reviewer 3 Report

The authors answered to my concerns and I can suggest acceptance of the paper in its present form. 

Author Response

Thank you for your answer and support.